# Adherence to clinical practice guidelines for South Australian pregnant women with cardiac conditions between 2003 and 2013

**Sandra Millington**[1]*, **Margaret Arstall**[2], **Gustaaf Dekker**[3], **Judith Magarey**[4], **Robyn Clark**[5]

**1** Adelaide Nursing School and Adelaide Medical School, Faculty of Health and Medical Sciences, University of Adelaide, Adelaide, South Australia, **2** Department of Cardiology at the Northern Adelaide Local Health Network (NAHLN) and the University of Adelaide, Adelaide, South Australia, **3** Obstetrics & Gynaecology for the Women's and Children's Division of Northern Adelaide Health Network (Lyell McEwin Hospital and Modbury Hospitals) and the University of Adelaide, Adelaide, South Australia, **4** Adelaide Nursing School, Faculty of Health and Medical Sciences, the University of Adelaide, Adelaide, South Australia, **5** College of Nursing and Health Science, Flinders University, Adelaide, South Australia

* sindy.millington@adelaide.edu.au

## Abstract

### Background

For pregnant women with a known cardiac condition or those who develop cardiac disease during pregnancy, there is an increased risk of complications during pregnancy, to both mother and foetus. To reduce this risk, best practice guidelines have been developed and available in South Australia for several years. Measuring clinical practice against the guideline recommendations verifies real-life practice and an essential part of any clinical practice quality improvement project by identifying gaps. This study is the first report on adherence to statewide perinatal guidelines for these women in South Australia.

### Objectives

- To evaluate adherence to evidence-based clinical practice perinatal guidelines
- To identify predictors of adherence.
- Make comparisons across three practice settings examined.

### Design

A retrospective cross-sectional observational design that analysed data from medical records.

### Setting

Three SA Health public metropolitan, university-affiliated teaching hospitals with an obstetric service within a ten-year timeframe (2003–2013).

**Data Availability Statement:** All relevant data are within the paper and its supporting information files.

**Funding:** The author(s) received no specific funding for this work

**Competing interests:** The authors have no competing interest

## Participants

271 admissions of women who were categorised as 'pre-existent' or 'newly acquired' cardiac condition during their pregnancy.

## Outcome measures

Adherence to guidelines was measured using a purposefully designed scoring system across the three sites. The researcher chose a minimum acceptable score of 17 applicable to the 'newly acquired' group and 35 for the 'pre-existent' group.

## Results

Overall adherence to the perinatal guidelines for the combined groups (n = 271) reported a mean score of 16.3, SD ± 6.7, with a median score of 17. Women in the 'newly acquired' group scored less compared to women in the 'pre-existent' group (Estimate -2.3, CI -3.9,-0.7). Variance in adherence was observed across the three hospitals (P value <0.0001). The most significant predictor of adherence to guidelines was pre-pregnancy cardiac consultation which increased the likelihood of preconception care by Odds ratio 18.5 (95%, CI 2, 168). Similarly, compliance with mental health screening was associated with improved adherence to antenatal assessments (OR: 11.3(95% CI 4.7, 27.3).

## Conclusion

There was overall suboptimal adherence to the statewide guidelines for women with cardiac conditions in pregnancy. The variance in the level of adherence across the three hospitals correlated with the exposure to higher acuity cases, and that appropriate up- referral to a higher acuity hospital was intrinsically linked to better adherence. Recommendations include preconception counselling, and to ensure that all health practitioners have the skills, sufficient training and time to complete a comprehensive initial antenatal assessment

## Trial registration

ACTRN12617000417381

## Introduction

The epidemiology of cardiac disease in pregnancy varies internationally with 0.2–4% estimates; however, it remains under-researched in Australia as reported by Australian Maternity Outcome Surveillance System (AMOSS) [1, 2]. Cardiac disease in pregnancy encompasses a broad spectrum of conditions that include congenital heart disease (CHD), structural and aortic disease, cardiomyopathies, rhythm disorders and pregnancy acquired conditions such as ischemic heart disease (IHD) [3, 4]. Cardiac disease is associated with increased rates of morbidity and hospitalisation during pregnancy, with one in four women requiring admission, and is now one of the leading causes of maternal mortality [4–6]. These women have an increased likelihood of eclampsia, caesarean birth and postpartum haemorrhage [4]. Importantly, there are reportedly higher rates of unintended pregnancies in women with cardiac disease, which raises questions about health literacy, contraceptive knowledge and access to preconception and pregnancy counselling by appropriately qualified healthcare practitioners [7]. Evidence-

based and best practice guidelines help to define the current known optimum quality of clinical care for maternal and newborn health; therefore, adherence to these best-practice standards is a measurable objective for quality improvement efforts [8].

## Background and rationale

Clinical practice guidelines provide clinicians with updated evidence to maintain consistency and accelerate best practice [9]. Where obstetric guidelines were agreed upon and implemented, there has been a significant improvement in the clinical outcomes, as reported in Sweden and the Netherlands [9, 10]. The Registry of Pregnancy and Cardiac disease (ROPAC) established to study pregnant women with structural heart disease reported that adherence to guidelines, such as prepregnancy assessment, counselling, and interventions, was an essential factor in reducing mortality and cardiac deterioration [11].

Australian national guidelines for the care of healthy pregnant women published online in 2018 provide recommendations that support high quality, safe prenatal care in all settings; however, there were no specific guidelines for women with cardiac conditions during pregnancy [12,13]. Recent research which examined women's perception of prenatal care, provider adherence to antenatal care guidelines, the risk of pregnancy complications, and guidelines associated with cardiovascular risk factors reported suboptimal adherence [12, 14–16]. Given the increased risk of cardiovascular complications during pregnancy, with consequences for both mother and foetus, we need to identify the gap between existing evidence and clinical practice.

The South Australian Perinatal Practice Guidelines (SAPPGs), implemented in 2010, represented the Australian practice guidelines at the time and included those specific to cardiac disease in pregnancy [17]. While regular revisions occur, this is the first evaluation on the uptake of the statewide guidelines across the public health sector. Measuring clinical practice against the best practice is a crucial step to address the evidence-practice gaps [18]. This study aimed to evaluate three public hospitals' adherence to the guidelines, with the following three objectives:

1. Evaluate the adherence to the perinatal guidelines for women with pre-existent and newly acquired heart conditions during pregnancy.

2. Identify predictors of adherence for these women.

3. Make comparisons across the three hospitals examined.

## Methods

### Study design and setting

The study design was a retrospective cross-sectional observational study, using data collected from a comprehensive medical record review. All women who had been admitted for their obstetric care to one or more of the three SA Health public metropolitan, university affiliated teaching hospitals with an obstetric service between 2003 and 2013.

**Setting.** The three hospitals in this study represent the complete SA Health (Government funded) metropolitan public hospitals inpatient obstetric service, as described in the guidelines [17]. Hospital One provided intermediate care (level five) that encompassed tertiary maternal services, maternal cardiac and intensive care and specialised neonatal care services excluding babies less than 32 weeks. The quaternary centre, hospital Two provided (level six) care that included maternal cardiac, cardiothoracic surgical and intensive care, and neonatal intensive care services. Hospital Three provided a neonatal intensive care unit but no maternal cardiac or intensive care services.

## Participants and selection criteria

**Inclusion criteria.** The participants in the study were women with 'pre-existent' or 'newly acquired' heart disease during pregnancy. The 'pre-existing' group included women with both structural and electrical cardiac conditions before conception. Structural cardiac conditions included cyanotic and acyanotic congenital heart disease, a previous history of hypertensive, valvular and cardiomyopathic heart disease as well as those women with ischemic heart disease. Electrical cardiac conditions included genetically inherited channelopathies as well as the full spectrum of acquired bradyarrhythmias and tachyarrhythmias. Women with symptomatic benign ectopic beats were excluded from this group. Similarly, women in whom undiagnosed congenital cardiac conditions were unmasked during pregnancy were included in the 'newly acquired group'.

The 'newly acquired' group included women with pregnancy-associated cardiac events.

This includes any new symptomatic sustained arrhythmia, the unmaking of underlying prior cardiac conditions previously unrecognised, and new onset cardiac disease such as acute coronary syndrome, cardiomyopathy with or without decompensated heart failure, and infective endocarditis during pregnancy or up to 12 months postpartum. These inclusion criteria were similar to those used in the United Kingdom survey of obstetric deaths [19].

The International Statistical Classification of Diseases and Related Health Problems, Tenth Revision, Australian Modification (ICD-10-AM), identifies conditions related to or aggravated by pregnancy, childbirth or the puerperium (maternal or obstetric causes). The codes for this project were O99, O90.3, O89, O88, and I51, I42 and I25.5.

**Exclusion criteria.** Women excluded from the study did not have 'pre-existent' or 'newly acquired' heart conditions during pregnancy up to 12 months post-partum and specifically amniotic fluid embolism and non-cardiac related pulmonary embolus were excluded as 'newly acquired' heart conditions.

## Guideline adherence variables

All pregnant women had the prescribed standard care in their pregnancy record in addition to specific perinatal guidelines [17]. A scoring system was devised that measured adherence to available protocols and guidelines in this study [20,21]. The forty guidelines adherence variables used in this study [S1 Table] were equally weighted, giving a maximum score of forty to measure adherence. Positive documentation of the guidelines, regardless of the entry point, achieved a score.

Shared minimum guideline adherence variables are the routine antenatal care expected for both cardiac groups that include lifestyle risk assessment factors, such as smoking, weight, height and body mass index (BMI), prescribed and illicit drug use, medical comorbidities and foetal risk assessments [17]. Completion of the centre of perinatal excellence (COPE) national mental health screening tools was a shared guideline adherence variable. These tools include the Antenatal Psychosocial Risk Questionnaire (ANRQ) for depression and the Edinburgh Postnatal Depression Scale (EPSD), with both scores entered in the pregnancy record.

**Predictor variables.** The predictors for adherence to the guidelines analysed were hospital sites, the two categories of cardiac disease (pre-existent or acquired heart conditions), the risk level of pregnancy, collaboration with multidisciplinary teams, and maternal, perinatal factors.

The Perinatal predictor variables analysed were; gestational age at delivery, mode of delivery, live baby weight, birth weight percentile, length, and head circumference at birth, and Apgar scores. Anthropometric measures of newborn, i.e. length at birth and head circumference correlate with live baby weight as a simple, practical method for detecting 'small for gestational age' (SGA) and therefore were chosen as more reliable surrogate factors for foetal risk assessment where birthweight was less than 10[th] percentile [17].

## Data sources/measurement

The medical record review utilised an online data abstraction tool (DAT) by Auditmaker, developed from the statewide clinical guidelines for cardiac disease in pregnancy, version 2.0 [16]. Clinicians reviewed the feasibility of the DAT [S1 File]. Further validation and testing of inter-rater reliability occurred by comparing the data collected from four medical records by the researcher and research supervisor. There was 100% agreement in the data collected.

The data collection from medical records occurred at the three hospitals over two years from December 2014 to July 2016. The data were extracted from the women's pregnancy record, labour and anaesthetic charts, medical and nursing notes, and clinical pathways documentation. The study included a final number of 271 medical records for the statistical analysis (see Fig 1).

**Selection bias.**   The data collected from the three hospitals excluded private hospitals. Whenever cross-sectional studies occur exclusively in hospital settings, there is the potential for 'admission bias', and so the women studied do not reflect the population of SA [22]. The maternal and neonatal services available at each of the three hospitals may influence the interpretation of results.

**Sample size and power calculations..**   As this type of study was unique, the sample size and power were calculated using current data from pregnant women who had a documented plan of care. This calculation to determine the required sample size indicated that a sample of 196 patients would have $> 99\%$ power to detect a difference of 13% of patients with a care plan. This number would provide a margin of error of 0.02 and $\alpha$ set at 0.05 (Gold standard is 99%, expected was 86%). The final sample size of 271 provided a substantial 'margin for error' to compensate for missing data in some patient's notes.

**Sampling.**   The sample was data abstracted from all the medical records that met the inclusion criteria within the timeframe between 2003 and 2013.

## Statistical methods

The analysis was completed using both SPSS statistics Version 26 and Statistical Analysis System (SAS 9.4, SAS Institute Inc., Cary, NC, USA). Descriptive statistics of the overall adherence score and the two cardiac groups included frequency tables of the relevant categorical data and some text variables. Continuous data variables were identified as normally distributed on inspection of the histogram. Initially, the Pearson Chi-Square test was used to determine the relationship between hospitals and various variables and tabulated in a contingency table that reported frequencies, column percentages, and Chi-Square P values. A one-way analysis of variance (ANOVA) calculated a p value for continuous variables normally distributed on inspection of a histogram. The continuous variables not normally distributed on examination of the histogram, the Kruskal-Wallis test calculated a P value. Logistic regression models determined the significant predictors utilising the software, the Statistical Analysis System (SAS 9.4, SAS Institute Inc., Cary, NC, USA). The *p* value for all analyses was set at $< 0.05$ or 95% CI to consider statistical significance.

Univariate logistic regressions identified the factors for adherence to the guidelines[S4 Table]. Multivariable logical regression examined the correlation between various predictors and adherence to the guidelines[S5 Table]. Multivariable models constructed by the following method: for each outcome, predictors with p value $< 0.05$, and the two broad cardiac categories and hospitals were in the initial multivariable model. Where suspected collinearity was evident between several predictors (such as parity and gravida), the most significant were included. Using backward elimination, the predictor with the highest p value was removed, and the model rerun until all predictors had p value $<0.05$; except the two broad cardiac

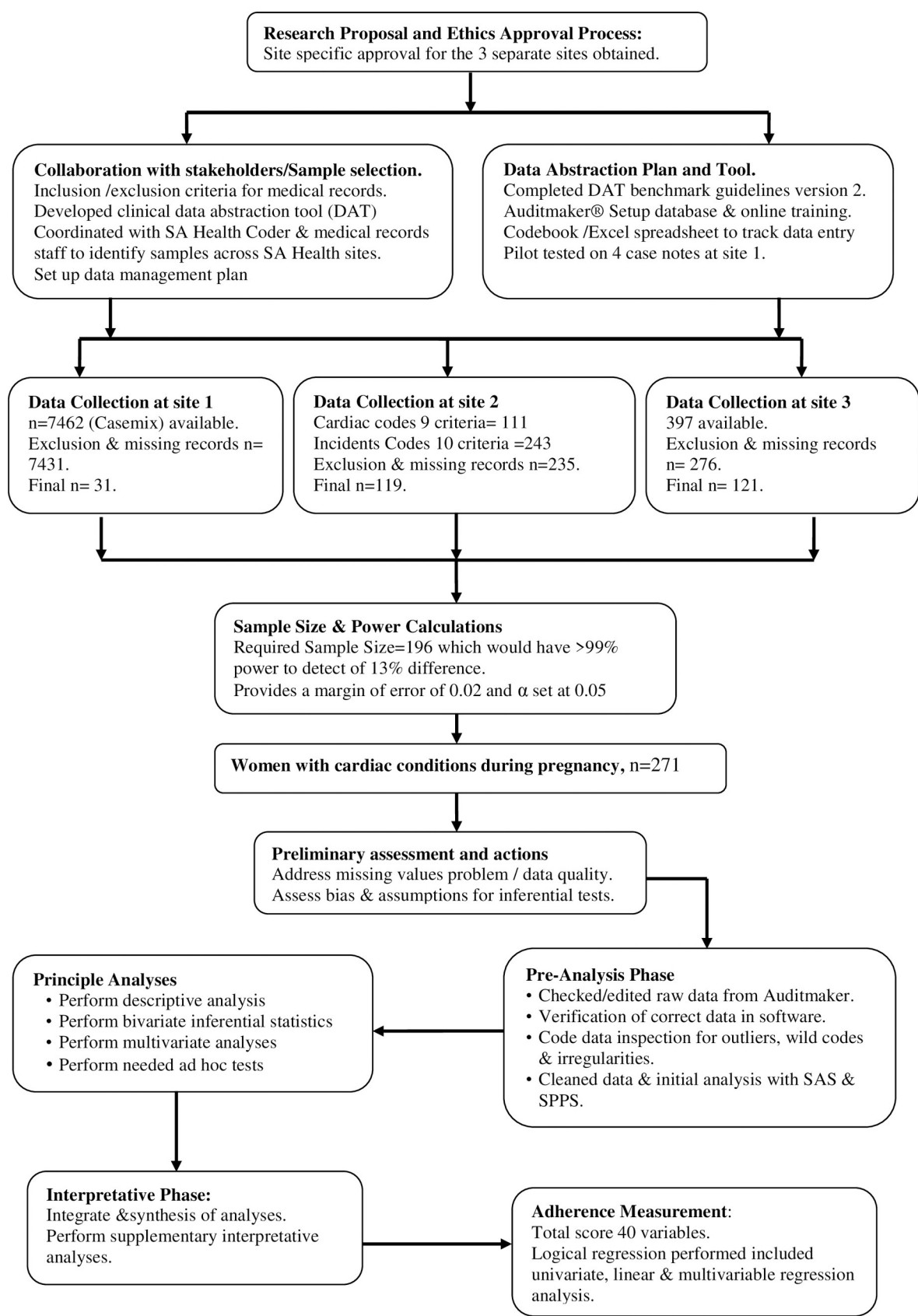

**Fig 1. Research flow chart.**

categories and three hospitals, which are included as priori predictors. The analysis yielded odds ratio for greater or lesser adherence score for content of the guidelines such as preconception, antenatal and planned care, anaesthetic and pain management during birth with individual factors that may have influenced the uptake of guidelines. Assumptions underlying these tests, such as the absence of collinearity, confounders and goodness of fit, were assessed [S2 Table].

## Ethics approval

South Australian Health Human Research Ethics Committee approved this study (Reference HREC 13 TQEH/LMH 226: Extension to Approval 03/08/2015). Separate site-specific ethics approvals were obtained from the three hospitals. Consent was not required from the participants for medical record review as the application for Low and Negligible Risk (LNR) Research Ethics approval granted the researcher access to medical records to collect data. To ensure anonymity, the participants' identifiable information was coded and stored with a separate password protected file on the university server and labelled for deletion in 15 years.

## Results

Maternal characteristics are detailed in Table 1. For the total cohort of women, most were born in Australia (81%), Caucasian (78%), married (61%), and lived in Adelaide metropolitan vicinity (77%) (Table 1). The mean age was 30 ±6years (SD). Multigravida women were predominant with $3 \pm 2.1$ pregnancies in the entire group. The mean parity of the women was 1.6 (SD ± 2.1) with 36% primiparous and 7% multiparous (parity $\geq$ 5 live or stillborn over 20 weeks gestation).

Over half the women ('pre-existent group', n = 143, 53%) had experienced a cardiac event during previous pregnancies. On examination of the cardiac conditions in pregnancies, arrhythmias (n = 76, 28%) and CHD (n = 62, 23%) were the most frequent causes. Rheumatic heart disease (RHD) (n = 36, 13%), heart failure (n = 29, 11%), pulmonary embolus in cardiac patient (n = 27, 10%), IHD and systemic arterial hypertension with severe cardiac conditions (equally n = 10, 4%) were less frequently reported.

### Adherence to the statewide perinatal guidelines

Overall adherence to the guidelines for the combined groups (n = 271) was normally distributed with a mean score of 16 ± 6.7 and a median score of 17. There was a significant association between the total score and the two cardiac groups (p = 0.001). The 'pre-existent' group reported a higher mean score of 17± 6.5 compared to the 'newly acquired group' with a mean score of 15± 6.8. Only eleven women had a documented NYHA classification, noting that the criterion did not apply to women with a new cardiac event. Both cardiac groups showed suboptimal adherence to the COPE screening tools. The ANRQ for depression reporting a mean adherence score of 18 ±13 while the EPSD a mean adherence score of 6±5.

### Predictors of adherence and comparisons across practice settings

Table 2 shows the estimates (and 95% CI) for comparison of the mean adherence scores with various medical and demographic predictors of adherence. Those predictors identified as not significant (Global p >0.05) were maternal age, ethnicity, social status, country of birth, gravida, gestational age and specific cardiac conditions such as RHD, arrhythmias, ischaemia, heart failure, and cardiac arrest, and therefore not presented in the table.

**Table 1. Maternal characteristics of women with cardiac disease during pregnancy SA hospitals (2003–2013).**

| Patient Characteristics | Frequency(Percentage) | Mean ± SD | | |
|---|---|---|---|---|
| | Totaln = 271 (100%) | Total Cardiac | Pre-existing cardiac (PEC) (n = 143) | Acquired cardiac (AC) (n = 128) |
| **Anthropometrics** | | | | |
| • Age (years) | 271 | 30.4 ± 6. | 29.8±5.7 | 31.2±6.3 |
| • Weight (kg) | 245 | 76.3 ± 19.6 | 73.2 ±17.8 | 79.9±21.0 |
| • Height (cm) | 186 | 165.1±8.0 | 165.1 ±8.5 | 165.0±7.3 |
| • BMI (kg/m$^2$) | 201 | 28.3±7.5 | 26.9±6.5 | 30.1±8.4 |
| †**Social parameters** | | | | |
| • Born in Australia | 221 (82) | | 116(81) | 105 (82) |
| • Born overseas | 50 (18) | | 27(19) | 23 (18) |
| †**Ethnicity** | | | | |
| • Aboriginal | 34 (13) | | 22(16) | 12(9) |
| • Asian | 15 (6) | | 8(6) | 7 (6) |
| • Caucasian | 213 (79) | | 109(77) | 104(82) |
| • African continent | 5 (2) | | 1(1) | 3(3) |
| • Other | 1 (0) | | 1(0) | 0(0) |
| †**Location of Home** | | | | |
| • Metropolitan | 208 (77) | | 103 (72) | 105 (84) |
| • Rural | 47(17) | | 31(2) | 17(14) |
| • Remote | 12 (4) | | 9(6) | 3(2) |
| †**Marital status** | | | | |
| • Married | 167(64) | | 84(60) | 83 (31) |
| • De facto | 53 (20) | | 34 (24) | 19 (7) |
| • Divorced /Separated | 5 (2) | | 2 (1) | 4(3) |
| • Single | 35(13) | | 19(4) | 15(12) |
| **Peri-natal assessment** | | | | |
| • Gestational age at admission | 265 | 36.8 ± 3.7 | 36.8±3.71 | 36.8±3.7 |
| • *Gravida | 268 | 3.1 ± 2.1 | 3.1 ±2.0 | 3.1 ±2.2 |
| • *Parity | 269 | 1.6 ±1.7 | 1.5 ±1.6 | 1.6±1.9 |
| • Blood pressure | | | | |
| ○ Systolic | 265 | 117.4 ± 17.8 | 117±17.6 | 117±18 |
| ○ Diastolic | 265 | 72.3. ± 12.6 | 72.2±14.2 | 72.4±14.6 |
| **Mental Health Scores** | | | | |
| ○ **ANRQ | 110 | 17.4. ± 12.6 | 15.7±12.0 | 19.6 ±13.2 |
| ○ ***$^E$PSD | 106 | 6.1 ±5.7 | 5.7±6.1 | 6.68 ±5 |

| Cardiac Characteristics | Total | PEC | AC |
|---|---|---|---|
| ○ Cardiac event (prev preg) | 144(54) | 103(72) | 0 |
| ○ Cardiac event (current preg) | | | |
| ○ Arrhythmia | 74(23) | 25(21) | 49(45) |
| ○ CHD | 55(24) | 53(45) | 2(2) |
| ○ RHD | 22(10) | 22(19) | 0 |
| ○ Heart Failure | 40(18) | 12(10) | 28(10) |
| ○ † Cardiac PE | 1(0) | 0 | 1 (1) |
| ○ IHD | 20 (9) | 2(2) | 18(17) |
| ○ Hypertension. | | 29 (10) | 29(10) |

*Gravida on admission = number of times a woman is or has been pregnant, regardless of the outcome. Parity = number of pregnancies reaching viable gestation age (includes live births & stillbirths).

**ANRQ = documented score for Antenatal Risk Questionnaire, self-reported psychosocial assessment tool.

***EPDS = documented score for the Edinburgh Postnatal Depression Scale for risk of perinatal depression with high score indicative of depression. †For these categorical variables only frequency and percentage are reported, see S5 Table.

†Cardiac PE = Cardiac pulmonary embolus.

Increased body mass index (BMI) was the only anthropometric maternal characteristic that resulted in a small but statistically significant increase in the mean adherence to guideline score.

Of the social characteristics where the woman's home location was important with rural and remote home location predicting a significantly higher mean adherence to the guideline score.

Women with a 'newly acquired' heart condition was a predictor of a lower score in comparison to women with 'pre-existent' heart issues with an estimate of -2.3, (95% CI -3.9, -0.7). Furthermore, a cardiac event in a previous pregnancy predicted a significant increase in adherence to the guidelines was reduced with an estimate of -1.7(95% CI -3.3, -0.1). Where the mode of delivery was planned adherence to the guidelines for caesarean options scored higher than vaginal delivery with an estimate of 6, (95%, CI: 5, 8). The baby characteristics in Table 2 did not directly influence adherence to guideline score, but retrospectively reflect the effect of this adherence. The lower mean adherence score was associated with smaller babies with lower Apgar scores

**Comparison across the three hospitals.**    Table 2 highlights a variance in adherence to the guidelines across the three hospitals (p <0.0001). An expected finding of increased adherence observed at hospital Two, which is the high-risk referral centre for other hospitals, including from remote and rural locations. Hospital Two adherence to the guidelines was greater in

**Table 2.  A comparison of the mean adherence score versus various 'predictors' for all women with cardiac conditions during pregnancy.**

| Predictor | Comparison | Effect on Total Guideline Adherence score Mean (95% CI) | Comparison P value | Global P value |
|---|---|---|---|---|
| Anthropometrics | | | | |
| ○ BMI | | 0.1 (0.02, 0.3) | | 0.03 |
| Social parameters | | | | |
| ○ Location of home | Metropolitan vs Remote | -5.0 (-8.8, -1.2) | <0.01 | <0.001 |
| | Metropolitan vs Rural | -3.1 (-5.2, -1.0) | 0.003 | |
| | Remote vs Rural | 1.9 (-2.2,6.0) | 0.36 | |
| Cardiac characteristics | | | | |
| ○ Onset of cardiac disease recognition. | Acquired vs Pre-existent | -2.3 (-3.9, -0.70) | | 0.004 |
| ○ Cardiac event in previous pregnancy | No vs Yes | -1.7 (-3.3, -0.1) | | 0.04 |
| ○ Congenital heart condition. | No vs Yes | -1.9 (-3.9, -0.07) | | 0.04 |
| ○ Bradycardia event during pregnancy. | No vs Yes | 13.4 (0.4, 26.5) | | 0.04 |
| ○ Pulmonary embolus during pregnancy | No vs Yes | 2.7 (0.1, 5.4) | | 0.05 |
| Hospital site for delivery | 1 vs 2 | -3.4 (-5.9, -0.9) | 0.008 | <0.0001 |
| | 1 vs 3 | 0.5 (-2.0, 3.0) | 0.68 | |
| | 2 vs 3 | 3.9 (2.3, 5.6) | <0.0001 | |
| Documented planned mode of delivery. | Caesarean vs Vaginal | 6.0 (4.6, 7.5) | | <0.0001 |
| Baby characteristics | | | | |
| ○ Live baby weight per 100g increase | | -0.2 (-0.3, -0.17) | | <0.0001 |
| ○ Baby's length at birth | | 0.4 (-0.6, -0.2) | | <0.001 |
| ○ Baby's Head circ at birth | | -0.4 (-0.7, -0.1) | | 0.030 |
| ○ Apgar 1min | | 0.7 (-1.1, -0.2) | | .003 |
| ○ Apgar 5min | | -1.0 (-1.7, -0.3) | | 0.005 |

Significant at p<0.05. NB: includes only significant predictors. Site 1 = Tertiary hospital, Site 2 = Quaternary hospital, Site 3 = Stand-alone maternity unit

comparison to hospital Three (estimate 3.9, 95% CI: 2.3, 5.6, p <0.0001). The tertiary hospital One adherence was lower to the quaternary hospital Two with an estimate of -3.4 (95% CI -5.9, 0.9).

**Preconception and antenatal care.**    Table 3 provides the stratification of both the factors (A) and the priori predictors of cardiac groups and the three hospital sites (B) that contribute to the uptake of components of the guidelines. The priori predictors are included in the table for each outcome variable to compare the cardiac groups, and hospital sites examined. The overall significant findings were that a higher adherence guideline score was predicted if a preconception cardiac consultation occurred (OR 18.5: 95% CI 2.0, 168) with no difference seen between hospitals. If mental health screening was performed during the antenatal assessment, this predicted a significantly higher guideline adherence with OR: 11.3, (95% CI 4.7, 27.3). There was a small but significant variation between hospitals for utilisation of these mental health tools. Antenatal documentation of maternal anthropometrics showed higher adherence at hospital One with an odds ratio of 4.6, (95% CI 2.2, 9.5, p <0.0001) reflecting this hospital's overall patient profile with a high incidence of obesity.

Women identified as a high risk pregnancy, with a documented plan of care, were more likely to have an echocardiogram assessment of their cardiac status (OR 2.6, 95% CI 1.1, 5.9) and (OR 9.4, 95% CI 3.1, 28) respectively.

**Management in labour.**    Guideline recommendations for thromboembolism prophylaxis are the use of anti-embolic stockings, administration of subcutaneous low molecular weight heparin (LMWH) or intravenous unfractionated heparin [17]. Prophylactic heparin administration was similar between the three hospitals; however, the greatest adherence was associated with the performance of cardiac consultation for management of pregnancy (OR: 65, 95% CI 3.6, 1171). Paediatric neonatal staff likelihood of attending the birth increased when a paediatrician and medical officer were already present (OR: 16.3, (95% CI 5.8, 45.7) and 7.2 (95% CI 2.3, 22.4) and p value <0.001.

**Anaesthetic and pain management during labour.**    The combined use of spinal-epidural anaesthesia recommended during labour has a positive effect on women with cardiac lesions [17]. When the multidisciplinary team were present at delivery and a plan was in place there was an increased likelihood of combined use of spinal–epidural anaesthesia (OR: 9.6, 95% CI 1, 81) (see Table 3).

## Discussion

This study to our knowledge is the first to evaluate adherence to the SA perinatal guidelines for women with cardiac conditions during pregnancy. A minimum score of acceptable guideline adherence was determined after a comparison of the two groups mean, and median adherence scores and expert review of selected cases identifying minimum expected care [S1 Table]. From this analysis, a score greater than 35 (for 'pre-existent') and 17 (for 'newly acquired') cardiac conditions were deemed acceptable guideline adherence.

Given the foreknowledge of the women's cardiac state, the minimum guideline adherence variables for the 'pre-existent group' would include preconception education, comprehensive antenatal assessment including a prepregnancy cardiac functional assessment using New York Heart Association (NYHA) classification and multidisciplinary team collaboration [14].

Conversely, in the 'newly acquired' group with a pregnancy-induced cardiac event, adherence to the guidelines will be contingent upon the timing of the cardiac event during the peripartum period. Furthermore, it is reasonable not to expect 100% concordance with the guidelines. The expected minimum guideline adherence variables would include adult

**Table 3. Significant factors that influenced the adherence score for the guidelines for cardiac disease in pregnancy across three public hospitals (2003–2013).**

| Adherence Guidelines Variable[1] | Factors for the uptake of guidelines. | | OR for adherence score (95% CI) | | Comparison P value | Global P value |
|---|---|---|---|---|---|---|
| | Predictor | Comparison | | | | |
| **Prenatal care:** Preconception care included<br>• Education & counselling for the cardiac condition during pregnancy. | A. Cardiac consultation completed.<br>• Gravida | Yes, vs No | 18.5 | (2.0, 168.8) | | 0.009 |
| | | | 1.6 | (1.03, 2.5) | | 0.04 |
| | B. Cardiac Condition | Acquired vs Pre-existent | 0.4 | (0.1, 1.8) | | 0.24 |
| | • Hospital Site for delivery | 1 vs 2 | 0.1 | (0.01, 2.1) | 0.15 | 0.28 |
| | | 1 vs 3 | 0.09 | (0.00, 1.8) | 0.12 | |
| | | 2 vs 3 | 0.8 | (0.2, 3.0) | 0.70 | |
| **Initial Antenatal Assessment i.e.**<br>• Weight, height & BMI. | A. Documented ANRQ. | | 11.3 | (4.7, 27.3) | | <0.0001 |
| | • Gestational age | | 1.0 | (1.0, 1.2) | | 0.02 |
| | B. Cardiac Condition | Acquired vs Pre-existent | 1.0 | (0.6, 1.8) | | 0.99 |
| | • Hospital Site for delivery | 1 vs 2 | 0.2 | (0.06, 0.6) | 0.003 | < .0001 |
| | | 1 vs 3 | 0.8 | (03, 2.3) | 0.69 | |
| | | 2 vs 3 | 4.6 | (2.2, 9.5) | <0.0001 | |
| • Oral health, respiratory & breast assessment. | A. Documented ANRQ | Yes, vs No | 0.9 | (0.9,0.97) | | 0.001 |
| | • Baby's length birth | Yes, vs No | 0.9 | (0.8, 1.0) | | 0.03 |
| | • Identified as high risk | Yes, vs No | 3.4 | (1.2, 10.3) | | 0.03 |
| | B. Cardiac Condition | Acquired vs Pre-existent | 2.0 | (0.7,5.5) | | 0.21 |
| | • Hospital Site for delivery | 1 vs 2 | 1.8 | (0.56, 5.8) | 0.30 | 0.10 |
| | | 1 vs 3 | 0.5 | (0.1, 1.7) | 0.24 | |
| | | 2 vs 3 | 0.3 | (0.07,0.9) | 0.03 | |
| • Abdominal, vaginal assessment & urinalysis. | A. - Identified as high risk | Yes, vs No | 1.7 | (1.0, 2.8) | | 0.04 |
| | • A documented plan of care | Yes, vs No | 2.6 | (1.2, 5.8) | | 0.02 |
| | B. B. Cardiac Condition | Acquired vs Pre-existent | 1.2 | (0.73, 1.99) | | 0.52 |
| | • Hospital Site for delivery | 1 vs 2 | 1.4 | (0.54, 3.41) | 0.63 | |
| | | 1 vs 3 | 1.1 | (0.43, 2.7) | 0.86 | |
| | | 2 vs 3 | 0.8 | (0.47, 1.34) | 0.39 | |
| **Mental Health Assessment:** Documented Antenatal Risk Questionnaire Self-reported (ANRQ) (for depression). | A. Documented Edinburgh Postnatal Depression Scale score | Yes, vs No | 75 | (16.9, 333.1) | | <0.0001 |
| | • Live baby weight | | 1.0 | (1.00,1.00) | | 0.01 |
| | • Cardiac Condition | Acquired vs Pre-existent | 0.4 | (0.2, 0.8) | | 0.01 |
| | B. Hospital Site for delivery | 1vs 2 | 0.7 | (0.1, 3.5) | 0.66 | 0.003 |
| | | 1 vs 3 | 2.5 | (0.5, 12.4) | 0.24 | |
| | | 2 vs 3 | 3.7 | (1.7, 7.9) | 0.001 | |
| **Antenatal/Ongoing Foetal Wellbeing assessment:** Fortnightly ultrasound or Growth, Doppler, & †AFI & CTG in the third trimester | A. Identified as high risk | Yes, vs No | 4.3 | (1.6, 11.7) | | |
| | • Pre-planned cardiac drugs in labour | Yes, vs No | 3.8 | (1.4, 10.5) | | 0.004 |
| | B. Cardiac Condition | Acquired vs Pre-existent | 1.0 | (0.4, 2.4) | | 0.01 |
| | • Hospital Site for delivery | 1 vs 2 | 0.6 | (0.09, 3.5) | 0.52 | 0.95 |
| | | 1 vs 3 | 2.4 | (0.5, 12.8) | 0.3 | 0.03 |
| | | 2 vs 3 | 4.4 | (1.5 12.6) | 0.007 | |

*(Continued)*

**Table 3.** (Continued)

| Adherence Guidelines Variable[1] | Factors for the uptake of guidelines. | | OR for adherence score (95% CI) | | Comparison P value | Global P value |
|---|---|---|---|---|---|---|
| | Predictor | Comparison | | | | |
| **Cardiac Echocardiogram** | A. Identified as high risk | Yes, vs No | 2.6 | (1.1, 5.9) | | 0.03 |
| | • A documented plan of care in place | Yes, vs No | 9.4 | (3.1, 28.2) | | <0.0001 |
| | B. Cardiac condition | Acquired vs Pre-existent | 0.5 | (0.2, 1.2) | | 0.12 |
| | • Hospital Site for delivery | 1 vs 2 | Not significant | | | 0.6 |
| | | 1vs 3 | Not significant | | | |
| | | 2 vs3 | 1.5 | (0.6, 3.40 | 0.33 | |
| **Management in Labour:** Prophylactic subcutaneous LMWH or IV unfractionated heparin. | A. Cardiac consultation for management of pregnancy & labour. | Yes, vs No | 65.3 | (3.6, 1171.7) | | 0.004 |
| | • Parity | | 0.4 | (0.2, 0.8) | | 0.01 |
| | B. Cardiac Condition | Acquired vs Pre-existent | 0.03 | (0.00, 0.07) | | 0.03 |
| | • Hospital Site for delivery | 1vs 2 | Not significant | | | 0.86 |
| | | 1vs 3 | Not significant | | | 0.86 |
| | | 2 vs 3 | 0.6 | (0.06, 5.2) | 0.60 | 0.86 |
| **Anaesthetic & Pain Management during labour.** • Components, i.e. anaesthetist consultation +epidural & spinal or alternative therapies. | A. Live baby weight per 1kg increase. | | 0.5 | (0.3, 0.7) | | <0.0001 |
| | • Obstetrician at delivery | Yes, vs No | 2.8 | (1.4, 5.7) | | 0.004 |
| | • Anaesthetist at delivery | Yes, vs No | 2.8 | (1.2, 6.7 | | 0.02 |
| | B. Cardiac Condition | | 0.4 | (0.2, 0.7) | | <0.001 |
| | • Hospital Site for delivery | 1vs 2 | 1.1 | (0.5, 3.0) | 0.76 | 0.04 |
| | | 1vs 3 | 0.6 | (0.2, 1.4) | 0.20 | |
| | | 2 vs 3 | 0.5 | (0.3, 0.9) | 0.01 | |
| Anaesthetist consultation for pain management in labour. | A. Live baby weight per 1kg increase. | | 0.5 | (0.3, 0.8) | | 0.004 |
| | • Identified as high risk | Yes, vs No | 2.6 | (1.3, 5.2) | | 0.001 |
| | • Anaesthetist at delivery | Yes, vs No | 2.8 | (1.0, 7.3) | | 0.03 |
| | B. Cardiac Condition | Acquired vs Pre-existent | 0.4 | (0.2, 0.8) | | 0.008 |
| | • Hospital Site for delivery | 1vs 2 | 1.3 | (0.4, 3.9) | 0.7 | 0.2 |
| | | 1vs 3 | 0.7 | (0.2,2.0) | 0.47 | |
| | | 2 vs 3 | 0.5 | (0.3, 1.0) | 0.06 | |
| • Epidural considered. | A. Apgar 1min | | 1.4 | (1.2, 1.6) | | < 0.001 |
| | • Anaesthetist at delivery | Yes, vs No | 6.8 | (2.9, 16.2) | | <0.0001 |
| | B. Cardiac Condition | Acquired vs Pre-existent | 0.5 | (0.2, 1.0) | | 0.05 |
| | • Hospital Site for delivery | 1vs 2 | 1.5 | (0.4, 5.9) | 0.53 | 0.68 |
| | | 1vs 3 | 1.1 | (0.3, 4.3) | 0.86 | |
| | | 2 vs 3 | 0.7 | (0.3, 1.7) | 0.45 | |
| • The combined use of epidural & spinal to decrease preload & reduce afterload. | A. Documented plan of care in place | Yes, vs No | 9.6 | (1.1, 81.0) | | 0.03 |
| | • No local anaesthetic with adrenaline used | Yes, vs No | 3.9 | (1.2, 12.8) | | 0.02 |
| | • Obstetrician at delivery | Yes, vs No | 3.6 | (1.07, 12.4) | | 0.04 |
| | • Medical officer at delivery | Yes, vs No | 2.2 | (1.03, 4.7) | | 0.04 |

(*Continued*)

**Table 3.** (Continued)

| Adherence Guidelines Variable[1] | Factors for the uptake of guidelines. | | OR for adherence score (95% CI) | | Comparison P value | Global P value |
|---|---|---|---|---|---|---|
| | Predictor | Comparison | | | | |
| | • Anaesthetist at delivery | Yes, vs No | 11.6 | (1.4, 94.2) | | 0.02 |
| | B. Cardiac Condition | Acquired vs Pre-existent | 1.4 | (0.7, 3.0) | | 0.35 |
| | • Hospital Site for delivery | 1 vs 2 | 0.2 | (0.04, 1.2) | 0.07 | 0.20 |
| | | 1 vs 3 | 0.3 | (0.05, 1.3) | 0.10 | |
| | | 2 vs 3 | 1.1 | (0.5, 2.5) | 0.84 | |
| • Inhalational gases, a general anaesthetic or alternative therapies. | A. Live baby weight per 1kg increase | | 0.6 | (0.4, 0.9) | | 0.02 |
| | • Identified as high risk | Yes, vs No | 2.0 | (1.0, 3.9) | | 0.03 |
| | B. Cardiac Condition | Acquired vs Pre-existent | 0.5 | (0.3, 1.0) | | 0.03 |
| | Hospital Site for delivery | 1 vs 2 | 0.9 | (0.3, 2.70) | 0.87 | 0.72 |
| | | 1 vs 3 | 0.7 | (0.2, 2.1) | 0.55 | |
| | | 2 vs 3 | 0.8 | (0.4, 1.5) | 0.47 | |
| Paediatric neonatal staff Present at delivery. | A. Live baby weight per 1kg increase | | 0.4 | (0.2, 0.9) | | 0.04 |
| | • Paediatrician at delivery | Yes, vs No | 16.3 | (5.8, 45.7) | | <0.0001 |
| | • Medical officer at delivery | Yes, vs No | 7.2 | (2.3, 22.4) | | <0.001 |
| | B. Cardiac Condition | Acquired vs Pre-existent | 0.6 | (0.2, 1.7) | | 0.4 |
| | Hospital Site for delivery | 1 vs 2 | 0.8 | (0.1, 5.2) | 0.82 | 0.78 |
| | | 1 vs 3 | 1.2 | (0.2, 7.0) | 0.82 | |
| | | 2 vs 3 | 1.5 | (0.5, 4.8) | 0.48 | |

Significant at P < 0.05. A = Factors for the uptake of guidelines. B = Priori Predictors that is cardiac group categories (Prexistent and Acquired) and the three hospital sites. Site 1 = Tertiary hospital, Site 2 = Quaternary hospital, Site 3 = Stand-alone maternity unit applies to all outcomes.

*ANRQ: Antenatal Risk Questionnaire Score for depression, self-reported psychosocial assessment.

**EPDS score: documented score for the Edinburgh Postnatal Depression Scale, 10 item questionnaire to identify women at risk of perinatal depression.

†Amniotic Fluid Index (AFI) and Cardiotocograpghy (CTG).

cardiologist or physician management soon after the cardiac event and the multidisciplinary team involvement for the rest of the peripartum care.

The study showed an overall suboptimal adherence to the guidelines. As expected, adherence was higher in women with 'pre-existent' heart disease. The most likely explanation would be that clinicians' foreknowledge of a woman's heart condition facilitates increased awareness and planning for potential complications during pregnancy [23, 24].

Stokes *et al.* [25], stated that "we need to understand better why guideline implementation strategies work in some contexts and not in others" and therefore, the variance observed across the three hospitals requires further explanation. The results show that hospitals which have increased exposure to higher acuity cases have improved adherence to the guidelines. Hospital Two is the only high-risk referral centre for SA and therefore, has increased exposure to high cardiac risk pregnancies. The patient selection will drive by default adherence to perinatal guidelines. There also appeared to be an increased awareness of the perinatal guidelines. In recognised complex cases, inserted in the women's case notes were the relevant sections of the guidelines, a delivery plan and speed dial numbers of team members. This strategy kept all clinicians informed of the expected care to facilitate an uncomplicated delivery.

Women identified as high-risk pregnancies in rural and remote locations received comprehensive assessment and investigations before transfer to hospital Two. Women transferred from rural and remote regions present a selection bias, as we were unable to determine the management of other obstetric patients from these locations. International and statewide guidelines recommend that women with moderate or high-risk complications during pregnancy require management and delivery coordinated at an expert centre with a multidisciplinary team [1].

There was documented cooperation between the three hospitals for high risk pregnancies, where women who required advanced care were identified for interhospital transfer [17]. Multidisciplinary team meetings did not occur at all three hospitals. However, high-risk physicians were prompt to refer women who experienced a pregnancy-induced cardiac event to a specialist following their thorough assessment. Hospital One had better adherence to the guidelines than hospital Three, which had limited maternal services so that women who required additional cardiac investigations may need to be transferred to a non-obstetric public hospital.

Elkayam *et al.* [24] emphasised that all women with cardiac disease benefit from pre-conception counselling, which provides a detailed discussion of the risk of pregnancy, and includes a comprehensive history and physical assessment. In this study, early cardiology consultation increased the likelihood of preconception care, particularly at hospitals One and Two. Women from regional, remote and interstate locations generally received preconception care before admission.

In previous studies, retrospective investigations of cardiovascular maternal mortality in pregnancy were, due to a missed diagnosis of a cardiovascular condition or new-onset cardiovascular disease as a common theme [26]. Wolfe *et al.* [27] stressed the importance of early cardiovascular screening for symptoms of CVD and improved management of hypertension as imperative to prevent maternal deaths. Barriers to pre-pregnancy assessment missed opportunities to identify cardiac risk factors during prenatal care, gaps in high-risk intrapartum care and delayed recognition of cardiovascular symptoms were contributory factors [27]. A notable incident in this study occurred when a pregnant primigravida woman attended the women's assessment clinic for blood pressure monitoring. Her clinical condition rapidly deteriorated so that she required advanced life support and a perimortem caesarean in the clinic. Post-cardiac arrest investigations revealed that the woman did not disclose to the midwifery staff her history of paediatric cardiac surgery for CHD, nor that she had no cardiology follow-up with the transition to adulthood. Importantly, routine physical examination, which is no longer included in the antenatal assessment, may have raised questions about the visible sternal scar on the woman's chest. Recent research highlighted inconsistencies in adherence to guidelines for the first antenatal care visit, with longer timeframes required to complete recommended prenatal risk factors screening, cited as a potential barrier [28]. Clinical time constraints and a woman's reluctance to disclose can culminate in abbreviated visits that exclude ideal health education [12, 28].

Antenatal assessments encompassing both the physical and psychological aspects of healthcare had better adherence at hospital One. This higher compliance with the COPE screening tools may be the result of the well-established perinatal mental health services in the hospital's family clinic [29]. Previous research described pregnant women attending the hospital One as among the most socio-economic disadvantaged group, with a high level of exposure to domestic violence during pregnancy, with a history of previous physical and emotional abuse during their childhood, placing them at high risk for mental illness in adult life and especially in the peripartum and postpartum period [30]. Ongoing collaborative research projects such as the Health-e Babies App for antenatal education help embed a research culture within the hospital [29]. Hospital One also had improved adherence to pain management (see Table 3). A likely

explanation could be a more holistic approach with established pain management regimen and interdisciplinary (obstetric and anaesthetic departments) collaboration that prioritises pain relief for vulnerable women from the surrounding socio-economic community.

## Factors that influence the uptake of guidelines

Although it was not feasible to objectively measure the barriers and facilitators identified in previous research in this retrospective study, three broad categories resonated in this project. The three broad categories of external barriers, patient and health care team factors were considered pertinent and are presented in Fig 2 [9, 23, 31].

Although recent research reported a disparity in maternity services for Australians living in rural and remote areas, interesting results were observed in this study [32]. Health care professionals in these postcode locations may have an increased risk awareness, initiate early multidisciplinary collaboration and retrieval of the women to the appropriate maternal centre. It is important to consider the fact that the study results only reflect those women referred to a metropolitan hospital, while the indigenous women dying in remote locations are typically poorly registered, and by default not captured in this type of hospital-based research. An example in this study of the lack of physical resources was the explanation provided for the omission of a requested 12 lead electrocardiograph (ECG) due to the inability to source a functional ECG machine, a situation not unique to Australia [19].

Nair *et al.* [33] identified patient barriers such as reduced health literacy, lack of opportunities for shared decision making, and confidence in healthcare providers. In this study, women frequently did not take the advice offered by the health care team, failed to arrive for appointments and arrived in labour, having had no prenatal care throughout their pregnancy.

**Fig 2. The factors that may influence the uptake of perinatal practice guidelines for women with cardiac conditions during pregnancy.**

Therefore, the resulting low adherence to guideline score could not be attributed to the control of the healthcare team.

Although perinatal staff assist in the development process and revisions of the guidelines, not all health care providers are familiar with the guidelines. Barth *et al.* [34] reported that clinicians lack familiarity with guidelines, agreement with the content, or have a deficit in the necessary skills to deliver care were all barriers to the uptake of guidelines [33]. Staff prioritised documentation of perinatal data for labour and the delivery that is reported to the birth registry. High-quality documentation and extraction of clinical information, as well as dynamic clinical leadership, have been identified as enablers to improve implementation strategies for guidelines [25].

## Conclusion

Overall, there was a suboptimal adherence to the SA statewide guidelines for women with cardiac conditions in pregnancy. The actions undertaken comprehensively before or early in the pregnancy resulted in ongoing adherence to the guidelines. The observed variance in the level of adherence across the three hospitals reflected the exposure to higher acuity cases, and that appropriate referral was intrinsically linked to better adherence. The early inclusion of the multidisciplinary team facilitated adherence to the guidelines, mainly with preconception care, and echocardiogram assessment of cardiac status.

The authors propose that perinatal guidelines should contain realistic and clear recommendations, which allow individual clinical judgement orientated for the patient as for efficacy versus safety. Two pivotal recommendations are the completion of preconception counselling and a comprehensive initial antenatal assessment. Therefore, it is vital to ensure that all health practitioners have the skills, training and sufficient time to complete the initial assessment [3].

### Generalizability

The data collection was limited to South Australia Health public hospitals due to access and availability of data from the medical records. The sample size and findings are not reflective of the women in South Australia.

### Limitations

A limitation of this study is generalizability with the data collected from three South Australian metropolitan public hospitals and private hospitals excluded. This study does not reflect state or national population proportions of women with cardiac conditions in pregnancy. Our study encountered the limitations associated with retrospective medical records reviews, such as missing data and accuracy of documented care delivered. The evidence of adherence to guidelines and the quality of data was contingent upon the quality of documentation. Clinicians may have provided care; yet there was, no documentation in the case notes. The researcher carefully examined individual case notes for other evidence of the care provided in the case notes.

## Supporting information

**S1 Table. Supplementary information code log for auditmaker and SAS scoring for adherence to the statewide perinatal guidelines.**
(PDF)

**S2 Table. Assumption testing.**
(PDF)

**S3 Table. Linear regressions.**
(PDF)

**S4 Table. Univariate regression.**
(PDF)

**S5 Table. Descriptive statistics cardiac variables.**
(PDF)

**S1 File. Perinatal audit DAT.**
(PDF)

## Acknowledgments

The authors acknowledge and are grateful to all staff in the Medical Records Department of the three sites for their invaluable support. We would also like to express appreciation to the statistician Suzanne Edwards, from the University of Adelaide, for her invaluable assistance on statistics. Sandra Millington receives support through an Australian Government Research Training Program Scholarship. A Heart Foundation Future Leader Fellowship (APP ID. 100847) supports Robyn Clark.

## Author Contributions

**Conceptualization:** Sandra Millington, Margaret Arstall, Gustaaf Dekker, Judith Magarey, Robyn Clark.

**Data curation:** Sandra Millington.

**Formal analysis:** Sandra Millington.

**Funding acquisition:** Sandra Millington.

**Investigation:** Sandra Millington.

**Methodology:** Sandra Millington, Gustaaf Dekker.

**Project administration:** Sandra Millington.

**Resources:** Sandra Millington, Gustaaf Dekker, Judith Magarey.

**Supervision:** Margaret Arstall, Gustaaf Dekker, Judith Magarey, Robyn Clark.

**Validation:** Sandra Millington, Margaret Arstall, Gustaaf Dekker, Judith Magarey, Robyn Clark.

**Visualization:** Sandra Millington, Margaret Arstall, Gustaaf Dekker, Robyn Clark.

**Writing – original draft:** Sandra Millington.

**Writing – review & editing:** Sandra Millington, Margaret Arstall, Gustaaf Dekker, Judith Magarey, Robyn Clark.

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
