## [Decision Letter · Decision Letter 0]

3 Dec 2019

PONE-D-19-28739

Adherence to clinical practice guidelines for South Australian pregnant women with cardiac conditions between 2003 and 2013

PLOS ONE

Dear Mrs Millington,

Thank you for submitting your manuscript to PLOS ONE. After careful consideration, we feel that it has merit but does not fully meet PLOS ONE’s publication criteria as it currently stands. Therefore, we invite you to submit a revised version of the manuscript that addresses the points raised during the review process.

Please look that there are several suggestions that mus be faced and responded. We will evaluate a revised version of the manuscript.

We would appreciate receiving your revised manuscript by Jan 17 2020 11:59PM. To enhance the reproducibility of your results, we recommend that if applicable you deposit your laboratory protocols in protocols.io, where a protocol can be assigned its own identifier (DOI) such that it can be cited independently in the future. For instructions see: http://journals.plos.org/plosone/s/submission-guidelines#loc-laboratory-protocols

We look forward to receiving your revised manuscript.

Kind regards,

Ricardo Q. Gurgel, PhD

Academic Editor

PLOS ONE

Journal Requirements:

2. In ethics statement in the manuscript and in the online submission form, please provide additional information about the patient records used in your retrospective study. Specifically, please ensure that you have discussed whether all data were fully anonymized before you accessed them and/or whether the IRB or ethics committee waived the requirement for informed consent. If patients provided informed written consent to have data from their medical records used in research, please include this information.

Additional Editor Comments (if provided):

Reviewers' comments:

Reviewer's Responses to Questions

**Comments to the Author**

1. Is the manuscript technically sound, and do the data support the conclusions?

Reviewer #1: Partly

Reviewer #2: Yes

2. Has the statistical analysis been performed appropriately and rigorously? 

Reviewer #1: Yes

Reviewer #2: Yes

3. Have the authors made all data underlying the findings in their manuscript fully available?

Reviewer #1: No

Reviewer #2: Yes

4. Is the manuscript presented in an intelligible fashion and written in standard English?

Reviewer #1: No

Reviewer #2: Yes

5. Review Comments to the Author

Reviewer #1: The manuscript evaluated and discussed a relevant theme, however it is not clear. Introduction shows reference 9 and 10 as the guidelines providing recommendations but reference 9 is a study about following the previous guidelines. Objective number 2 is described as "identify predictors of the quality of care for these women" but in fact it evaluated guideline adherence. If it means quality of care must be justified. The hospital codes are reported in inclusion criteria, it would be more clear if described cardiac conditions. Exclusion criteria that are showed are unnecessary. Hospital 1, 2 and 3 are described out of order and reading becomes confused. Scoring system should be an attached file to improve comprehension of the method. How did you choose the number 35 to classify as minimum acceptable score for 'pre-existent' cardiac condition and 17 for 'newly acquired'? Sample was extracted from ALL medical records that met the inclusion criteria? You observed significative differences between pre-existent and newly acquired group but all predictors are described for all subjects. Tabela 3 is the most important in the manuscript but is not understandable, to much data. Study design evaluating guideline adherence by reviewing medical records has some limitations and they are commented but how to consider barriers to adherence in this model? The writing and tables (specially table 3) must be reviewed to improve the study comprehension.

Reviewer #2: In the presente manuscript entitled "Adherence to clinical practice guidelines for South Australian pregnant women with cardiac conditions between 2003 and 2013", the authors investigate the adherence to evidence-based clinical pratice perinatal guidelines and make comparisons across three metropolitan hospitals, between 2003 e 2013. The authors conclude that there was overall suboptimal adherence to the statewide duidelines for women with cardiac conditions in pregnance.

General comments

This is a well-designed study that addresses a relevant topic using appropriate statistical analysis. The main limitation, as mentioned by the authors themselves, is the lack of comparison of findings with private institutions.

Minor comments

Abstract

Line 26: - The background was lacking, which justifies the research;

Introduction

Line 75: It was not clear to the reviewer which conditions are considered "Cardiac Diseases";

Inclusion Criteria

Line 119: Why in the "Pre-existing" group did the authors consider only women with cogenital pathologies, eventually disregarding those with hypertensive, ischemic, valvular and cardiomyopathic heart disease acquired before pregnancy?

Lines 121-124: The defining criteria for the "New Acquired" group are mixed: Acute Coronary Syndromes (ACS) include ST-segment elevation acute myocardial infarction, ST-segment elevation acute myocardial infarction, and unstable angina; therefore, ischemic herat disease (IHD) is a form of ACS; "Angina" refers to Chronic Coronary Syndromes ?; On the other hand, "Cardiomyopathies" acquired during pregnancy can be confused with pre-existing undiagnosed diseases. Do the authors refer to "Peripartum Cardiomyopathy"? And any "myocarditis"; the occurrence of "Pregnancy Specific Hypertensive Disease" was not considered; and finally "arrhythmia" has a broad meaning, encompassing a broad spectrum of gravity.

Outcome Variables

Line 149: Were the scores used in the present investigation validated for the Australian population?

Predictor Variables

Line 169: Why were not included in the "Maternal Characteristics", hemodynamic variables such "Systemic Blood Pressure" and "Heart Rate"?

Results

Line 285: What is the meaning of "Hypertension"? Do the authors refer to "Pulmonary Hypertension" or "Systemic Arterial Hypertension"?

Discussion

Line 400: I suggest including a comment of this type: We can speculate that Guidelines should contain realistic and clear recomendations, to be a room for individual judgement clinic and orientated for patient as for efficacy versus safety

6. PLOS authors have the option to publish the peer review history of their article (what does this mean?). If published, this will include your full peer review and any attached files.

Reviewer #1: No

Reviewer #2: Yes: Antônio Carlos Sobral Sousa, MD, PhD, FACC

---

## [Author Response · Author response to Decision Letter 0]

5 Feb 2020

I have provided a tabulated response to both reviewers to specific feedback in the attached documents. 

I have attached a copies of the initial ethics approval and subsequent extension (Not for publication) in response to the editor's query regarding to medical record access. The manuscript has included an ethics statement regarding participants consent for medical record review which was not required for the Low and Negligible Risk( LNR ) application for retrospective medical review at commencement of research. NHMRC Policy (2007) in place at the time of commencement of research has been provided and url to website below : NHMRC Policy 2007 

Section 2 Risk and Benefit, Consent. 

Chapter 2.1 Risk and Benefit Guidelines 

2.1.6 Research is 'low risk' where only the foreseeable risk is one of discomfort.. 

involvement in the research carries no more than low risk (see paragraphs 2.1.6 and 2.1.7, page 18) to participants

 2.17 Research is 'negligible risk' where there is no risk of harm or discomfort..page 18.

Chapter 2:3 Qualifying or Waiving conditions for consent Guidelines.

Limited disclosure 

2.3.1 (C) the research involves no more than low risk to participants (as above 2.1.6. pg. 18 ), and the limited disclosure is unlikely to affect participants adversely.

Waiver 2.3.6 

2.3.10 Before deciding to waive the requirement for consent (other than in the case of research aiming to expose illegal activity), an HREC or other review body must be

 satisfied that: 

a) involvement in the research carries no more than low risk (see paragraphs 2.1.6 and 2.1.7, page 18) to participants

 b) the benefits from the research justify any risks of harm associated with not seeking consent

 c) it is impracticable to obtain consent (for example, due to the quantity, age or accessibility of records) 

 d) there is no known or likely reason for thinking that participants would not have consented if they had been asked

 e) there is sufficient protection of their privacy

 f) there is an adequate plan to protect the confidentiality of data

page 24 

2.3.6(C) 2007

https://www.nhmrc.gov.au/about-us/publications/national-statement-ethical-conduct-human-research-2007-updated-2018#toc__235

---

## [Decision Letter · Decision Letter 1]

2 Mar 2020

Adherence to clinical practice guidelines for South Australian pregnant women with cardiac conditions between 2003 and 2013

PONE-D-19-28739R1

Dear Dr. Millington,

We are pleased to inform you that your manuscript has been judged scientifically suitable for publication and will be formally accepted for publication once it complies with all outstanding technical requirements.

With kind regards,

Ricardo Q. Gurgel, PhD

Academic Editor

PLOS ONE

Additional Editor Comments (optional):

Reviewers' comments:

Reviewer's Responses to Questions

**Comments to the Author**

1. If the authors have adequately addressed your comments raised in a previous round of review and you feel that this manuscript is now acceptable for publication, you may indicate that here to bypass the “Comments to the Author” section, enter your conflict of interest statement in the “Confidential to Editor” section, and submit your "Accept" recommendation.

Reviewer #1: All comments have been addressed

Reviewer #2: All comments have been addressed

2. Is the manuscript technically sound, and do the data support the conclusions?

Reviewer #1: Yes

Reviewer #2: Yes

3. Has the statistical analysis been performed appropriately and rigorously? 

Reviewer #1: Yes

Reviewer #2: Yes

4. Have the authors made all data underlying the findings in their manuscript fully available?

Reviewer #1: Yes

Reviewer #2: Yes

5. Is the manuscript presented in an intelligible fashion and written in standard English?

Reviewer #1: Yes

Reviewer #2: Yes

6. Review Comments to the Author

Reviewer #1: The manuscript highlights a relevant theme in clinical practice by a good quality research. The upgrade done after revision made it more clear.

Reviewer #2: In the presente manuscript entitled "Adherence to clinical practice guidelines for South Australian pregnant women with cardiac conditions between 2003 and 2013", the authors investigate the adherence to evidence-based clinical pratice perinatal guidelines and make comparisons across three metropolitan hospitals, between 2003 e 2013.

Only minor inquiries were found in the manuscript that were adequately answered by the authors. Therefore, I recommend the paper for publication.

7. PLOS authors have the option to publish the peer review history of their article (what does this mean?). If published, this will include your full peer review and any attached files.

Reviewer #1: No

Reviewer #2: Yes: Antônio Carlos Sobral Sousa

---

## [Editor Report · Acceptance letter]

5 Mar 2020

PONE-D-19-28739R1 

Adherence to clinical practice guidelines for South Australian pregnant women with cardiac conditions between 2003 and 2013 

Dear Dr. Millington:

I am pleased to inform you that your manuscript has been deemed suitable for publication in PLOS ONE. Congratulations! Your manuscript is now with our production department. 

With kind regards,

on behalf of

Professor Ricardo Q. Gurgel 

Academic Editor

PLOS ONE